# The Drying Kinetics and CFD Multidomain Model of Cocoa Bean Variety CCN51

**DOI:** 10.3390/foods12051082

**Published:** 2023-03-03

**Authors:** Eduardo Castillo-Orozco, Oguier Garavitto, Omar Saavedra, David Mantilla

**Affiliations:** 1Facultad en Ingeniería Mecánica y Ciencias de la Producción, Escuela Superior Politécnica del Litoral, ESPOL, Campus Gustavo Galindo Km. 30.5 Vía Perimetral, Guayaquil P.O. Box 09-01-5863, Ecuador; 2Center of Nanotechnology Research and Development (CIDNA), Escuela Superior Politécnica del Litoral, ESPOL, Campus Gustavo Galindo Km. 30.5 Vía Perimetral, Guayaquil P.O. Box 09-01-5863, Ecuador

**Keywords:** conjugate cocoa drying, computational fluid dynamics, drying technology

## Abstract

The CCN51 cocoa bean variety is known for being highly resistant to diseases and temperature variation and for having a relatively low cultivation risk for the producers. In this work, a computational and experimental study is performed to analyze the mass and heat transfer within the bean when dried by forced convection. A proximal composition analysis is conducted on the bean testa and cotyledon, and the distinct thermophysical properties are determined as a function of temperature for an interval between 40 and 70 °C. A multidomain CFD simulation, coupling a conjugate heat transfer with a semiconjugate mass transfer model, is proposed and compared to the experimental results based on the bean temperature and moisture transport. The numerical simulation predicts the drying behavior well and yields average relative errors of 3.5 and 5.2% for the bean core temperature and the moisture content versus the drying time, respectively. The moisture diffusion is found to be the dominant mechanism in the drying process. Moreover, a diffusion approximation model and given kinetic constants present a good prediction of the bean’s drying behavior for constant temperature drying conditions between 40 and 70 °C.

## 1. Introduction

Cocoa (*Theobroma cacao* L.) is a crop from humid and tropical lands, produced mainly by small farmers; this crop is sometimes grown without specialized techniques [1]. The lack of technical progress in artisanal production in developing countries means that postharvest processes could be more robust in most cases, especially in terms of the fermentation and drying of the cocoa seeds. The production of cocoa and its derivatives has increased continuously in the last decade, impacting socioeconomic development [2]. In Latin America, this product is mainly produced in Brazil, Colombia, Venezuela, Peru, and Ecuador [3]. Ecuador is one of this region’s largest producers, with an annual production of 360,000 t in 2021. This represents about 35% of the total exports from this country [3]. Ecuadorian cocoa is recognized worldwide for its aroma, flavor, and quality, and it is highly demanded in the European and North American markets [4].

The postharvest processes of fermentation and drying are essential to meet the expected acidity, flavor, and quality [5]. Thermal dehydration is also a weight reduction method and prevents the decomposition of the product.

The two main cocoa varieties produced in Ecuador are: (a) National Cacao, also known as Cacao Arriba. This type of bean is a wonderful indigenous Ecuadorian variety, which offers a smooth and aromatic chocolate [6]; and (b) Cocoa CCN51 (Castro Naranjal 51 Collection), which was developed through cloning. This variety is highly resistant to diseases and climate fluctuation and has a low cultivation risk for the producer. The CCN51 variety is the most produced in Ecuador and is used mainly to make chocolate in large volumes and to produce cocoa powder.

The CCN51 variety has become the most chosen alternative due to the low investment and production risk [6]. For this reason, the quality improvement of the bean in the postharvest process is essential, especially to compensate for the use of additives such as enzymes and yeasts before fermentation, either through the same fermentation or through the suitable convection drying process [7].

The postharvest operations can be divided into [8]:The transformation of the cocoa seeds into cocoa beans. This includes pretreatment with chemical additives, fermentation, and drying.The production of chocolate from the bean. In this process, roasting is highlighted.

The chocolate flavor is influenced by postharvest operations such as the chemical pretreatment of the seed and the fermentation, drying, and roasting of the dried beans [9]. Biochemical compounds are produced during the fermentation process [10]. After fermentation, the drying process of CCN51 cocoa is the most important. The quality parameters, e.g., the organoleptic properties, improve considerably based on the different drying parameters.

Convective dehydration is the most used drying process. In this process, water is extracted from the fermented beans to leave a moisture content in the grain less than or equal to 7% (dry basis) to stop its fermentation and decomposition, reduce its mass, and increase its shelf life [10]. The drying behavior is influenced by a series of internal parameters, e.g., the density, permeability, porosity, sorption and desorption characteristics, thermophysical properties, and the external parameters such as the temperature, velocity, and relative humidity [11,12]. Other drying procedures are solar drying [13], hybrid solar drying [14], oven drying, microwave drying, and freeze-drying [15].

Others have studied this process, but a thorough drying kinetics study with the thermophysical characterization of the CCN51 cocoa variety has yet to be carried out. Lukinov [16] performed a theoretical analysis of this phenomenon. The results deduced a mathematical model for the drying behavior of capillary-porous bodies considering different parameters. However, a mathematical model specifically for cocoa drying is still being determined [17]. Chua et al. [18] analyzed the drying kinetics of jaboticaba berries experimentally using pretreatment to accelerate the drying process in convective drying and to increase the ratios of sugar and acid in the dehydrated fruit. The moisture diffusivity was decreased by approximately 27% after pretreating the fruit with 70% sugar and by 70% with a 10% salt solution. Wanderley et al. [19] studied the drying kinetics of pomegranate peels and seeds in a hot air circulation oven at different temperatures. Different mathematical models were fitted to the experimental data. The best results were the diffusion approximation and the Verma model for the peel data and the diffusion approximation and the modified Henderson and Parbis model for the seed data.

Computational analysis is also used in food processing for diagnostics [20] and optimization [21]. Irudayaraj et al. [22] used the finite element model to numerically model the drying of single soybean, barley, and corn kernels. Jha and Tripathy [21] developed a 3D multilayered finite element model for predicting the drying behavior of the paddy during a simulated hybrid solar drying process. The numerical results allowed the optimum parameters, i.e., the power, air velocity, and moisture content, to be deduced. Computational fluid dynamics (CFD) has also been used to simulate this phenomenon for two-dimensional (2D) axisymmetric and three-dimensional (3D) models of other fruits and vegetables [23]. Azmir et al. [24] used a computational fluid dynamics–discrete element model to predict the moisture reduction and volume shrinkage for wheat drying. The results showed that the shrinkage rate increased significantly with the increase in air temperature, but the air velocity did not have a significant effect. Kan et al. [25] developed an intermittent microwave convective drying (IMCD) framework by integrating a CFD model with the heat and mass transfer. This analysis indicated that CFD with an IMCM model affected the drying kinetics, as the heat and mass transfer coefficients varied spatially throughout a cylindrical apple sample.

There are three drying periods for cocoa beans: the constant drying rate period, the penetration falling rate, and the regular regime falling rate period [26,27]. Nganhou [28] determined temperature profiles and drying curves for cocoa beans from the region of Yaounde in Cameroon. Hii et al. [29] studied the kinetics of heat pump drying of cocoa beans from Malaysia. The results showed that the moisture reduction was relatively faster at the testa compared to the cotyledon in the initial two hours. Still, the moisture content of the cotyledon became lower than that of the testa. Additionally, experiments and numerical simulations were used to analyze the heat and mass transfer under a stepwise drying condition [30]. The results indicated that the shrinkage of the bean did not play a significant role in the drying kinetics.

This work compares different numerical approaches to identify an optimum CFD model to analyze the drying kinetics of CCN51 cocoa beans by forced convection. For this purpose, the specific thermophysical properties of the ecotype of cocoa CCN51 from Ecuador were determined, as a function of the temperature inside the beans, both for the testa and the cotyledon part. In this way, it was possible to study the phenomena of the heat and mass transport within these two well-differentiated continuums from the point of view of their biological microstructures. In addition, experimental data were used to validate the computational model that can subsequently be employed for predicting the drying curves under different drying conditions. Furthermore, we determined the driving mechanism of the phenomenon and an appropriate mathematical model for the drying behavior of this cocoa variety. This allows designers to use this model to optimize the current technical solutions for CCN51 cocoa dehydration, especially since drying experimentation is a long process that can take several hours.

## 2. Materials and Methods

### 2.1. Thermophysical Properties

A proximate composition analysis was conducted to determine the thermophysical properties of the Ecuadorian echo type of cocoa CCN51. The thermal properties of the sample, i.e., the density, specific heat, thermal conductivity, and the thermal diffusivity, were calculated using Choi’s equations [31]. The method is based on the product’s composition, specifically, the proteins, fats, carbohydrates, fibers, water, and ash content. The fermentation of the cocoa beans was carried out for 7 days, until 100 g of testa and 100 g of cotyledon were obtained. Figure 1 shows a peeled bean and its cross section depicting the testa (outer covering shell) and the cotyledon (embryonic part or core). These two major parts of the bean are macro- and microstructurally different; thus, the proximate composition was evaluated separately for the testa and the cotyledon. The humidity content was determined according to the INEN 1676 standard [32], the protein according to the AOAC 970.22 standard (%N × 6.25) [33], the ash according to the AOAC 930.30 standard [33], the fat according to the AOAC 2003.06 [33], and the carbohydrates by calculation. The results of the proximate composition of the bean are presented in Table 1.

The results shown in Table 1 were then used for predicting the temperature-dependent thermophysical properties of the major constituents of the bean by applying a polynomial model [34]. Next, the thermophysical properties of the testa and the cotyledon were determined based on the weight fraction and the thermal properties of the major pure components for an interval between 40 and 70 °C, i.e., the Choi and Okos’ [31] prediction model. The density, heat capacity, and thermal conductivity were determined as $\rho =1/\sum \left(X_{i}/\rho_{i}\right)$, $C_{p}=\sum \left(C_{pi}X_{i}\right)$ and $k=\sum \left(k_{i}V_{i}\right)$, where $X_{i}$ and $V_{i}$ are the mass and volume fraction of the ith food component. The specific thermal diffusivity was calculated from $\alpha =k/\left(\rho C_{p}\right)$. Table 2 shows the thermophysical properties of the sample at 50 °C. In addition, a second-degree polynomial curve fitting of the discrete data was used to realize the properties’ temperature dependence of the bean (Equations (1)–(6)). These temperature-dependent equations for the testa and the cotyledon were utilized within the CFD simulation to account for the temperature field within the bean domain.

Thermal properties of the shell:(1)$$\rho_{s}\left(\mathrm{kg}/m^{3}\right)=-0.0017\;{\left(T\left({}^{\circ}C\right)\right)}^{2}-0.2415\;T\left({}^{\circ}C\right)+1133.9,$$
(2)$$Cp_{s}\left(\mathrm{kJ}/\mathrm{kg}{}^{\circ}C\right)=-1\times 10^{-6}{\left(T\left({}^{\circ}C\right)\right)}^{2}+0.0011\;\;T\left({}^{\circ}C\right)+2.6096,$$
(3)$$k_{s}\left(W/m{}^{\circ}C\right)=-4\times 10^{-6}{\left(T\left({}^{\circ}C\right)\right)}^{2}+0.002\;\;T\left({}^{\circ}C\right)+0.3405.$$

Thermal properties of the core:(4)$$\rho_{c}\left(\mathrm{kg}/m^{3}\right)=-0.0032\;{\left(T\left({}^{\circ}C\right)\right)}^{2}-0.0633\;T\left({}^{\circ}C\right)+1146.9,$$
(5)$$Cp_{c}\left(\mathrm{kJ}/\mathrm{kg}{}^{\circ}C\right)=2\times 10^{-6}{\left(T\left({}^{\circ}C\right)\right)}^{2}+0.0006\;T\left({}^{\circ}C\right)+3.2591,$$
(6)$$k_{c}\left(W/m{}^{\circ}C\right)=-6\times 10^{-6}{\left(T\left({}^{\circ}C\right)\right)}^{2}+0.0017\;T\left({}^{\circ}C\right)+0.4766.$$

### 2.2. Experimental Drying Setup

Drying experiments were carried out to validate the computational model based on the bean temperature and the drying curve of the CCN51 cocoa bean. An individual cocoa bean was dried experimentally by forced convection at constant fluid temperatures of 50 °C. The bean was treated with 7 days of anaerobic and aerobic fermentation. The experiments were repeated 5 times for accuracy.

Figure 2 shows the experimental setup. A forced convection drying oven (RS-40P Electric Thermostatic Dry Box REBELK) was used (Figure 2a). Hot air was blown over the cocoa bean at a speed of 1 m/s. The air speed was measured with an anemometer (HHF803 anemometer, resolution 0.05 m/s). The measurements were made at a point located 6 cm before the sample. A special sample holder was used to suspend 2 cocoa beans in the air and expose them to the incoming hot air. Figure 2b shows the sample holder, where 2 cannulas worked as quick disconnects to allow the rapid temperature and mass measurements of the beans. This reduced the induced changes in the drying oven while the measurements were performed. One bean was held fixed in one of the cannulas, where a T-type thermocouple probe was placed inside the bean for temperature measurements at the center. This bean remained fixed to the sample holder during drying to measure the local temperature evolution versus the time. The second cannula was removable. Another bean was placed in this cannula to obtain its mass as a function of the drying time. The sample was weighed every 20 min during the experiment. Each time the cannula was disassembled, it was placed in a desiccator to prevent moisture reabsorption. At the same time, the mass measurement was carried out on an analytical balance of 0.001 g readability to gravimetrically detect small changes in the moisture content in the individual grain. After the dataset was obtained, the dry bone weight was determined using a thermobalance, drying the sample with infrared radiation at 70 °C until the mass of each grain did not vary. The average initial weight of the beans used in 5 experiments was 1.973 g, while it was 1.63 g when the beans were dried.

The ambient temperature and humidity were 26 °C and 72%, respectively. The air temperature inside the drying chamber was 50 °C. The detrimental temperature to cocoa quality is approximately 60 °C [35]. Dehydration at higher temperatures produces excessive acid entrapment inside the beans, which causes flavor loss.

### 2.3. Governing Equations and the Numerical Model

Two main models can numerically simulate the drying process: (1) a distributed model considering the simultaneous heat and mass transfer. This model considers the internal and external heat and mass transfer, predicting the temperature and the moisture gradient. This model depends on the Luikov equations that arrive from Fick’s second law of diffusion; (2) a lumped parameter model that does not consider the temperature gradient in the body, and it assumes a uniform temperature distribution that equals the drying air temperature in the product.

This study built a coupled CFD model with the mass transport equation inside the cocoa bean. Figure 3 outlines the experimental and computational procedure to solve the coupled problem. The Navier–Stokes equations were solved outside the bean. The system of coupled partial differential equations is stated as follows:(7)$$\nabla \cdot \vec{V}=0,$$
(8)$$\rho \frac{D\vec{V}}{Dt}=\rho \vec{g}-\nabla p+\mu \nabla^{2}\vec{V},$$
(9)$$\rho C_{p}\frac{\partial T}{dt}+\rho C_{p}\vec{V}\cdot \nabla T=\nabla \cdot \left(k\nabla T\right),$$
(10)$$\frac{\partial c}{dt}-\nabla \cdot \left(D_{\mathrm{eff}}\;\nabla c\right)=0,$$
where Equation (7) through Equation (10) are the vector form of the incompressible form of the continuity equation, the momentum equation, the energy equation, and Fick’s law equation, respectively.

To estimate the effect of the temperature of the drying process on the effective diffusion coefficient of water in a cocoa bean $D_{\mathrm{eff}}$, an Arrhenius function (Equation (11)) was used, and it was taken from the work by Hii et al. [30].
(11)$$D_{\mathrm{eff}}\left(m^{2}/s\right)=D_{o}\exp\left(\frac{E_{a}}{R\cdot T\left(K\right)}\right),$$
where the pre-exponential factor $D_{o}$ is 12 × 10^−4^ m^2^/s, the activation energy $E_{a}$ is −38,000 J/mol, the universal gas constant R is 8314 J/(mol.K), and the temperature, T, is in absolute scale.

The finite element method (FEM) was employed to solve the coupled system of differential equations. An axisymmetric 2D model was used to simulate the bean with an axial flow of hot air using COMSOL Multiphysics. The geometry of the CCN51 bean was approximated to an elliptical shape with a length and width of 22 mm and 12 mm, respectively. Additionally, the testa of the bean was modeled by a shell with a thickness of 0.55 mm.

A hybrid mesh was chosen, as can be seen in Figure 4. An unstructured mesh within the bean and throughout the fluid domain was used to generate a complex grid with larger elements. A standard grid size of 0.1 mm was used to create the shell mesh. A structured mesh and refinement were used for the boundary layer over the solid–fluid interface to capture the gradients in this region. Different mesh types were tested to check for grid independence. Finally, the temperature at the center of the solid domain (cocoa bean core) was examined and compared with that of the experimental measurements to evaluate the grid resolution effect. The final grid consisting of 142,000 elements was achieved once there was no significant variation in the transient temperature curve (Figure 5).

A symmetry boundary condition was used in the axisymmetric axis, i.e., thermal isolation (Equation (12)) and zero mass transfer (Equation (13)). At the inlet, a constant air velocity of 1 m/s and constant air temperature varying from 40 to 70 °C was set. The convective heat and mass transfer were considered at the solid–fluid interface and were obtained using Equation (14) and Equation (15), respectively. The initial moisture concentration was set at $c_{o}$ = 10,878 mol/m^3^ and the initial bean temperature at $T_{o}$ = 24 °C.
(12)n·(−k∇T)=0,
(13)n·(−D∇c)=0,
(14)$$n\cdot \left(-k\nabla T\right)=h\left(T_{\mathrm{air}}-T\right)+\lambda \left(D\nabla c\right),$$
(15)$$n\cdot \left(-D\nabla c\right)=h_{m}\left(c_{\infty }-c\right).$$

The numerical model was built by coupling a conjugate heat transfer model with a semiconjugate mass transfer model. This means that the Navier–Stokes and energy equations were solved simultaneously both in the fluid domain of hot air and in the solid domain (within the bean). With this approach, it was not necessary to specify a film convective coefficient at the solid–fluid interface, as one of the limitations of modeling only the solid domain [30] is that a given film coefficient h must be specified.

The mass transfer was only solved within the cocoa bean, making it necessary to specify a natural or Newman boundary condition (Equation (15)). This boundary condition consisted of the convective mass flow that was determined using the Sherwood analogy (Equation (16)) for a flat plate [29,36], where the Sherwood number ($S_{h}=Lh_{m}/D_{\mathrm{air}}$) is a function of the Reynolds ($R_{e}=LV_{\mathrm{air}}/\nu$) and Schmidt numbers ($S_{c}=\nu /D_{\mathrm{air}}$).
(16)$$S_{h}=2+0.552R_{e}{}^{0.53}S_{c}^{1/3}.$$

The heat transfer rate across the solid–fluid interface was coupled to the nonisothermal air flow; thus, the local rate of the heat transfer was dependent on the local fluid velocity and temperature; however, the flow field was assumed to be independent of the temperature and moisture content.

## 3. Results and Discussion

### 3.1. Experimental Validation of the Numerical Results

The temperature variation and the bean’s moisture content were compared quantitatively with the experimental results in Figure 5 and Figure 6, respectively. Figure 5 compares the numerical predictions with the experimental bean temperature as a function of the drying time under a constant temperature drying condition of 50 °C and a constant flow velocity of 1 m/s. It can be observed from the experiment that after approximately 1 h, an equilibrium temperature of 44 °C was reached inside the bean. Different mesh types and sizes were used to perform a grid convergence analysis. Some of these results are shown in Figure 5 to depict the final grid-independent CFD model used for the rest of this study (minimum cell size of 0.1 mm). Figure 6 presents the variation in the moisture content dry basis, which is given in units of g of water per g of dry matter as a function of the drying time for the same drying conditions. Similarly, the numerical prediction was compared with the experimental result. Here the multidomain model proposed in this work was compared with that of one single domain, showing that a multidomain approach was more accurate. The differentiated physical properties and the ability to use a more suitable mesh for each region yielded better results. However, it should be considered that using an additional core–shell interface increased the computation time. The results from the CFD simulation matched well with the experimental results. The average relative error in the bean temperature and the moisture content versus drying time was 3.5% and 5.2%, respectively.

Moreover, the velocity, temperature, and moisture concentration fields were obtained inside and outside the bean from the CFD simulations. The velocity field on the fluid flow domain and the temperature field and the moisture content distribution inside the bean at a time step that corresponded to 240 min after drying started are shown in Figure 7. It can be observed that a stagnation point was depicted on one side of the bean, and a wake of low velocity and recirculation was noted on the opposite side (Figure 7a). The bean reached an equilibrium temperature of 44 °C or 317 K (Figure 7b). In addition, this time step was sufficiently high; so, a moisture gradient was observed inside the bean. The maximum concentration was in the central zone (Figure 7c). Furthermore, the moisture concentration was calculated as the volume-weighed instantaneous average within the bean (shell and core). Thus, it was calculated for each time step during the CFD simulation. This methodology was used to reproduce Figure 6.

### 3.2. Drying Kinetics

Figure 6 displays the drying curve comparison between the computational and experimental results for a constant drying temperature of 50 °C. The curve shows the drying behavior based on the moisture content dry basis (Equation (17)) with the drying time. The experimental and numerical results started from the same moisture content on a dry basis, and they decreased exponentially within the first 240 min of drying. Moreover, the same CFD multidomain model could be used to predict the drying behavior at different drying temperatures. As the moisture content decreased, the bean size reduction or shrinkage was observed in the experiments. The shrinkage range was 8–12% for the drying experiments at 50 °C. However, it was decided not to incorporate it, as another numerical study [30] showed that using a shrinkage model did not have a significant effect on the simulated drying behavior of another type of cocoa bean. To find an appropriate universal drying model, the moisture ratio (Equation (18)) was used instead, where c is the moisture content, $c_{o}$ is the initial moisture content, and $c_{e}$ is the equilibrium moisture content.
(17)$$MC_{db}=\frac{\mathrm{wet}\;\mathrm{weight}-\mathrm{dry}\;\mathrm{weight}}{\mathrm{dry}\;\mathrm{weight}}$$
(18)$$MR=\frac{c-c_{o}}{c_{o}-c_{e}}$$

A diffusion approximation model (Equation (19)) [37] was fitted using nonlinear regression to the moisture ratio of the CCN51 cocoa bean with the drying time for the constant drying temperatures of 40, 50, 60, and 70 °C. The kinetic constant, $k_{o}$, and the drying constants, a and b, are presented in Table 3. Figure 8 shows the drying behavior of the bean under these drying conditions. Moreover, it was found to follow a diffusion model with a good degree of estimation, as observed. For all cases, the correlation of the coefficient was high (R^2^ >0.9999), and the root mean square was low (RMSE < 0.0044). Additionally, as the drying temperature increased, the mass transfer phenomenon occurred faster. This is explained in the sense that the effective moisture diffusivity increased with the temperature and directly influenced the drying rate. The moisture mass transfer during the drying process is a complex phenomenon involving various transport mechanisms, some of which are based on diffusion, such as vapor diffusion, surface diffusion, and molecular diffusion, and others, such as pressure differences [37]. Overall, the diffusion approximation (Equation (19)) and the constants listed in Table 3 presented a good prediction of the drying behavior for a single CCN51 cocoa bean.
(19)$$MR=a\exp\left(-k_{o}t\left(\min\right)\right)+\left(1-a\right)\exp\left(-k_{o}bt\left(\min\right)\right).$$

Figure 9 shows the moisture field within the bean at different times for a drying temperature of 50 °C. This sequence of images depicts the concentric gradients within the bean for what is represented in each point of Figure 8. It can also be observed that the moisture fields shifted slightly to the left within the grain. This was due to the influence of the direction of the external flow (from right to left in these images). In addition, the moisture concentration changed notably until 240 min (Figure 9a–c), while it remained similar for times between 240 and 350 min (Figure 9d–f), showing the approximation of an equilibrium moisture content. This equilibrium moisture content was calculated as 0.02 g of water/g of dry matter.

Finally, the multidomain model used in this study allowed the prediction of the drying kinetics at different conditions based on the parameters obtained experimentally and numerically. Moreover, it offered an alternative for qualitatively and quantitatively understanding the transport phenomena within the cocoa bean and the fluid flow domain.

## 4. Conclusions

A proximate composition analysis of the Ecuadorian echo type of cocoa, variety CCN51, was conducted using samples previously fermented for 7 days. In addition, the thermal properties of the testa and the cotyledon were determined as a function of the temperature for an interval between 40 and 70 °C. Furthermore, a computational multidomain 2D axisymmetric model was proposed and validated under a constant drying temperature of 50 °C. The bean temperature was used to evaluate the grid resolution effect. The grid independence was achieved within a hybrid mesh of 142,000 elements and refinement near the solid–fluid interface. The final CFD model worked well, predicting the transport phenomena within CCN51 cocoa beans, with average relative errors of 3.5% and 5.2% for the temperature and moisture content within the bean, respectively. The drying behavior for the constant temperature drying conditions at 40, 50, 60, and 70 °C was computationally predicted. Based on these results, a diffusion approximation model was suggested for the moisture ratio as a function of the drying time. The drying constants for this model were determined and listed in the results. Moreover, the numerical CFD model allowed the visualization of the velocity, temperature field, and moisture concentration within the bean to establish the drying parameters and kinetics of the drying process.

Although industrial drying equipment generally works with trays and forced convection, in-depth studies of the transport phenomena with one grain are a preliminary step to the optimization of the process parameters and the development of drying equipment. Therefore, further studies are recommended experimentally and through numerical simulation of a grain tray. The CFD simulation of grain trays remains challenging, mainly due to the meshing and computational consumption. Furthermore, the randomness of the position of the grains during the drying in the tray and the experiments only allows the simulation to make predictions but not to be deterministic.

## Figures and Tables

**Figure 1 foods-12-01082-f001:**
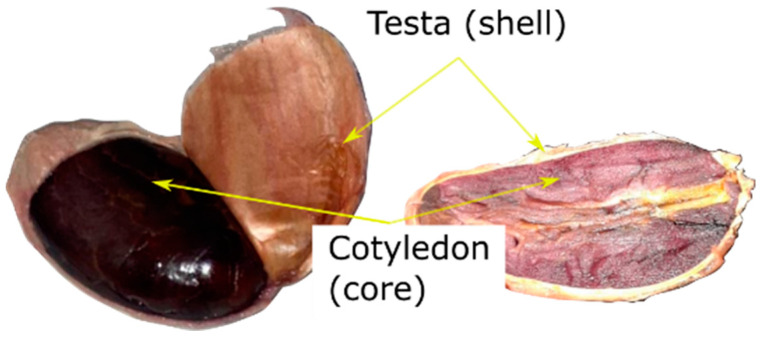
The fermented CCN51 cocoa bean. A cross-sectional view of the bean after 7 days of fermentation.

**Figure 2 foods-12-01082-f002:**
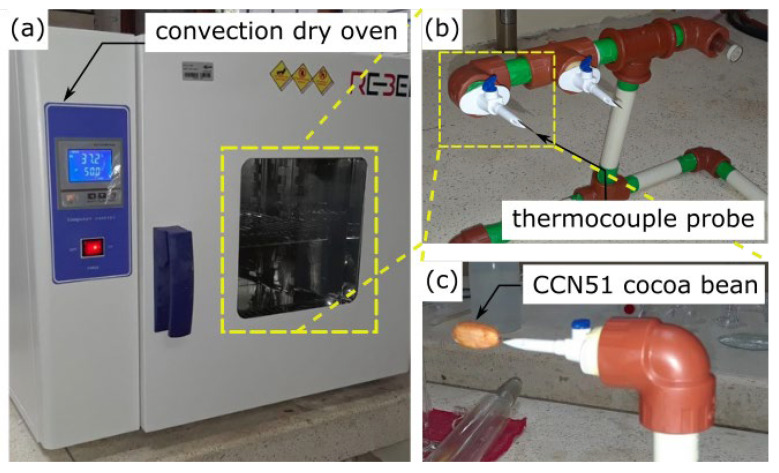
The experimental setup for the drying process of a cocoa CCN51 bean. (**a**) Convection dry oven; (**b**) sample holder; (**c**) bean inserted into a detachable cannula.

**Figure 3 foods-12-01082-f003:**
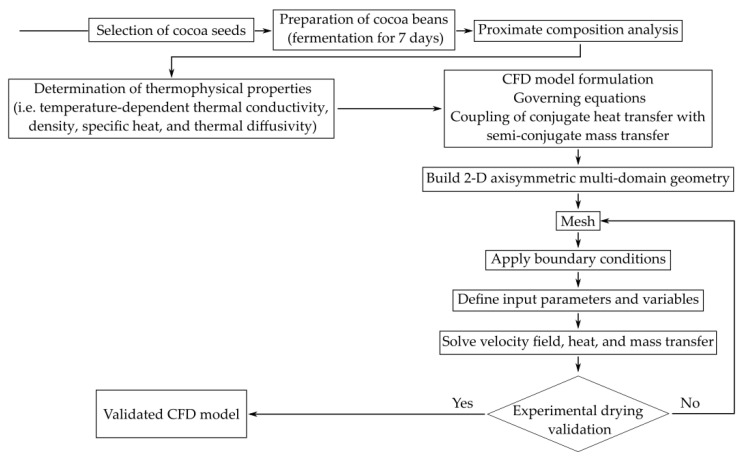
The overall experimental and computational procedure.

**Figure 4 foods-12-01082-f004:**
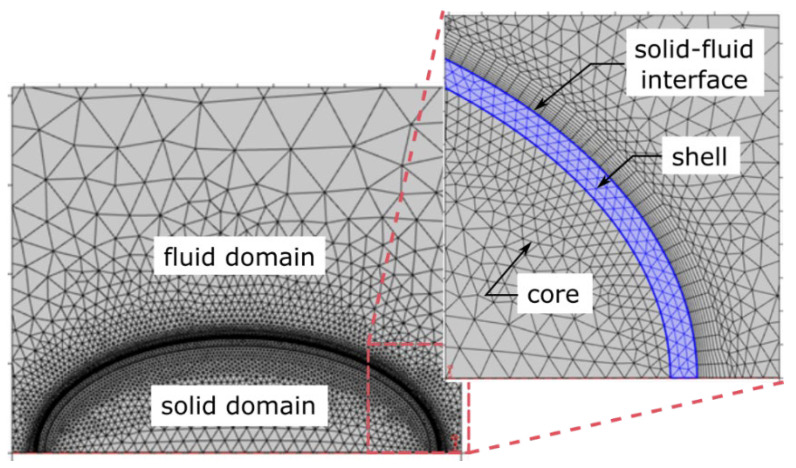
The computational domain and mesh. The 2D axisymmetric mesh of the solid domain, the solid–fluid interface, and the fluid domain. The solid domain includes the bean core and shell.

**Figure 5 foods-12-01082-f005:**
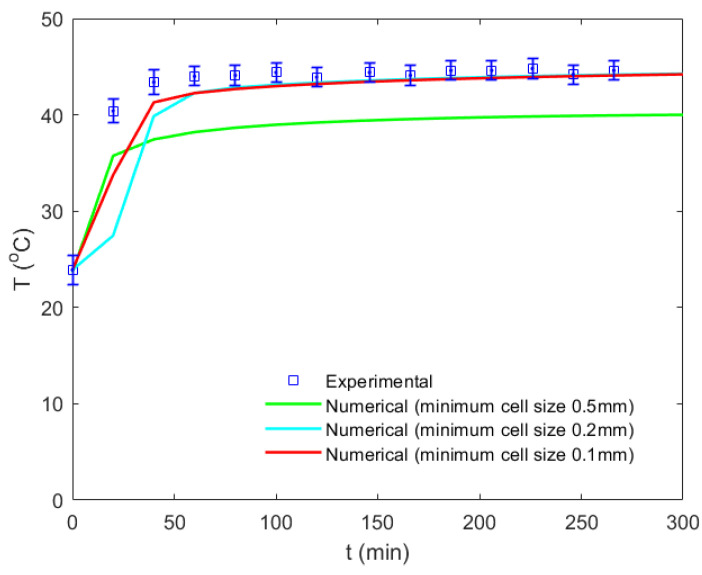
The grid convergence study based on the bean core temperature versus the drying time under a constant temperature drying condition of 50 °C.

**Figure 6 foods-12-01082-f006:**
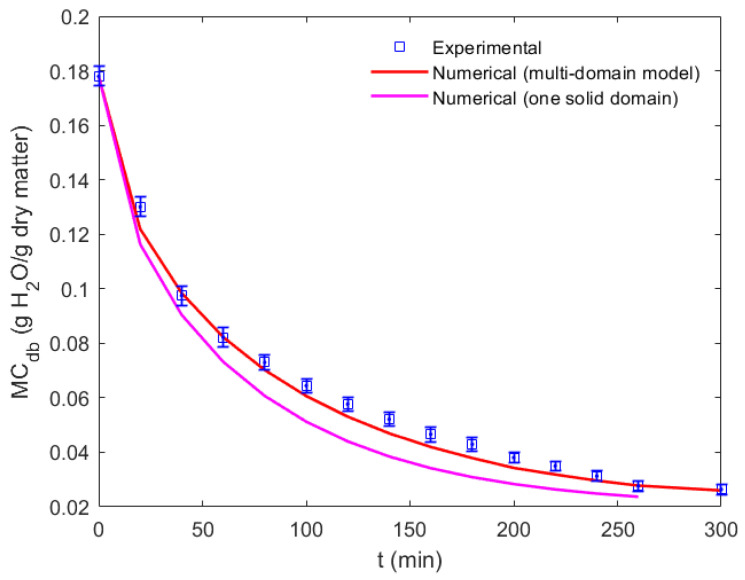
The moisture content dry basis versus the drying time. The quantitative comparison of the numerical results using one solid domain and a multidomain model (shell and core) with the experimental results under a constant temperature drying condition of 50 °C.

**Figure 7 foods-12-01082-f007:**
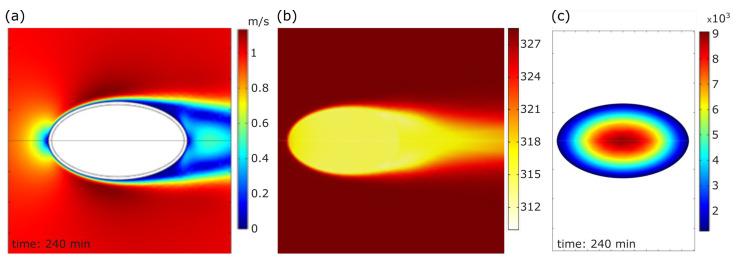
The computational velocity and temperature field and moisture concentration in the bean under a constant temperature drying condition of 50 °C at 240 min after drying started: (**a**) the velocity field (m/s) around the bean with a freestream flow velocity of 1 m/s; (**b**) the temperature field (K) in the solid (shell and core) and the fluid flow domain; (**c**) the dry basis moisture concentration field (mol/m^3^) inside the cocoa bean at 240 min.

**Figure 8 foods-12-01082-f008:**
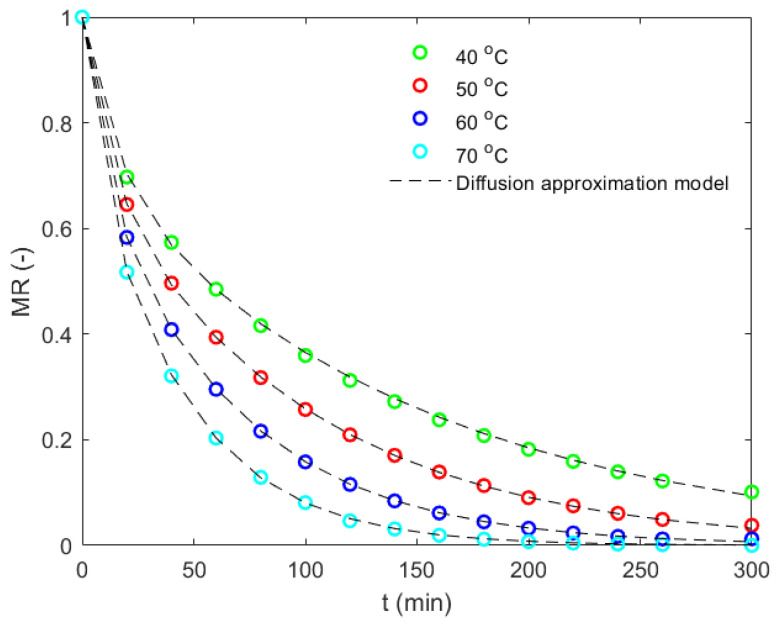
The computational drying kinetics of the CCN51 cocoa bean at the constant drying temperatures of 40, 50, 60, and 70 °C. A diffusion approximation model is fitted to the moisture ratio with the drying time.

**Figure 9 foods-12-01082-f009:**
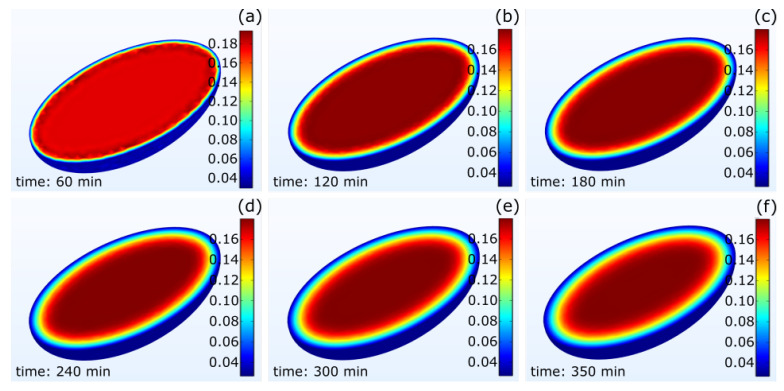
The time evolution of the dry basis moisture concentration (g H_2_O/g dry matter) inside the cocoa bean: (**a**) 60 min; (**b**) 120 min; (**c**) 180 min; (**d**) 240 min; (**e**) 300 min; and (**f**) 350 min after drying started.

**Table 1 foods-12-01082-t001:** The proximate composition of the Ecuadorian echo type of cocoa CCN51.

Sample	Protein (%)	Fat (%)	Carbohydrate (%)	Fiber (%)	Ash (%)	Water (%)
Testa (shell)	8.89	26.0	27.98	0.0	2.23	34.9
Cotyledon (core)	4.8	0.32	28.0	0.0	2.28	64.6

**Table 2 foods-12-01082-t002:** The thermophysical properties of the Ecuadorian echo type of cocoa CCN51 at 50 °C.

Sample	ρ (kg/m^3^)	$C_{p}$ (kJ/kg°C)	k (W/m°C)	α (m^2^/s) × 10^−7^
Testa (shell)	1117.6	2.660	0.427	1.436
Cotyledon (core)	1135.7	3.294	0.544	1.454

**Table 3 foods-12-01082-t003:** The drying constants for the diffusion approximation model at different temperatures from 40 to 70 °C.

Drying Constants	40 °C	50 °C	60 °C	70 °C
$k_{o}$ (min^−1^)	0.066	0.082	0.105	0.181
a	0.280	0.268	0.243	0.184
b	0.103	0.127	0.149	0.128

## Data Availability

Data is contained within the article.

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
