# Peer review of "The Drying Kinetics and CFD Multidomain Model of Cocoa Bean Variety CCN51"

_foods, 2023, doi:10.3390/foods12051082_

Round 1
Reviewer 1 Report
The abstract should be revised by reducing its length and improving its language.
Words such as they, their, and them should be avoided. Casual words such as "Generally speaking" in Line 174 should not be used. What is the basis of Equations 1-6? How these equations were derived? Elaborate. What is the value of hm in Equation 15? Specify. What is the value of h in Line 218? Specify. Some suitable reference should be cited for Line 288-290 "Moisture mass transfer during..". A flowchart for the research plan/design of the present study should be added. The following can be referred 10.1016/j.compag.2021.106215 The length of the article seems short. Literature on CFD analysis of grains should be presented in the Introduction section. Studies similar to the following must be cited 10.1007/s12393-020-09223-2 10.1016/B978-0-12-823696-3.00001-5 10.1016/j.compag.2021.106215 10.1007/s12393-022-09323-1
Reviewer 2 Report
The article "Drying Kinetics and CFD Multi-Domain Model of Cocoa Bean 2 Variety CCN51" has been reviewed and is considered to have been adequately developed and meets the requirements of a high-impact journal.
It is recommended to highlight in the summary and the introduction which are the most important contributions of the work and what is the impact in the sector that they want to influence.
A comparison was not clearly observed between previous works that have modeled the heat and mass transfer within foods that are subjected to drying processes. It is recommended to make a critical analysis of your work precisely to highlight the relevant aspects of it.
Additionally, a document is attached with more comments that will help improve the presentation of the work.

Reviewer 3 Report
A numerical study complemented by an experimental component concerning the drying kinetics of a cloned cocoa bean variety is described in the manuscript.
The manuscript does not deals with an original topic since modelling drying kinetics using numerical approaches with CFD is already being used extensively to simulate dehydration of foodstuffs. However, some novelty is noticed applying this of the CCN51 cocoa bean variety using a multi-solid domain (shell and core of cocoa bean) with distinct thermophysical properties. This can be viewed as an advancement of the knowledge in the field and useful to cocoa transformation industry to optimize drying stage operation conditions.
However, major corrections must be performed to reach the quality level deserving to be published.
Concerns/suggestions:
The thermophysical properties calculation after cocoa composition determination must be presented in more detail.
In Table 1 the number of significant figures must be presented adequately for all the parameters quantified. It must be also specified units of the values indicated (g per 100 g ?) and refer in the table legend “at 70 °C” makes no sense.
All values presented in Table 2 must be checked. For example, thermal diffusivity values are not well calculated. Why present thermophysical properties at 70 °C, if authors referred that cocoa exposed at temperatures higher than 60 °C suffers flavor losses?
How the air velocity over cocoa bean was measured inside the forced convection oven?
The (average) weight of the 5 cocoa beans used in the experiments must be indicate.
I have some concerns about the effect on the results of the methodology used interrupting every 20 minutes the drying process to weigh cocoa bean, even more in this case that a temperature probe must be located at the center of the solid every time … Additional comments must be presented. In fact, not only the conditions inside the forced convection oven changes but also the cocoa bean temperature that it will be used to validate numerical results.
Experiments were made at 50 °C and 60 °C but only data obtained at 50 °C were presented and used to validate numerical predictions.
Some comments about shrinkage of cocoa bean must be presented. Authors can certainly present this information with the individual cocoa bean dimension before and after drying.
Equation (16) is not used and for that reason is not necessary to present.
Emphasize the differences between the one single domain model and the multi-domain approach. Only different physical properties according the solid domain are used?
Text between lines 250 and 255: Indication of Figure 6 (b) at line 252 is not correct and the conclusion that comes from the fact that the maximum moisture content is in the central zone is not necessarily true.
In Figure 4 the predicted temperature profiles must start at 24 °C the initial bean temperature assumed.
The global moisture content evolutions in the bean cocoa obtained numerically and presented in Figure 5 certainly were calculated from the instantaneous moisture distributions inside cocoa bean. The methodology must be described.
Color scales in Figure 6 a) and b) is missing and in Figure 6 c) is not visible
The reference [30] used at line 279 seems not appropriate.
How the authors obtained the diffusion model parameters (Equation 20)? Using a non-linear regression?
The discussion of results must be improved in a scientific point of view.
Round 2
Reviewer 1 Report
The authors have made significant improvements. Now, the revised manuscript seems suitable for publication.
Author Response
We thank the Reviewer for taking the necessary time and effort to review our manuscript for a second time. We sincerely appreciate all his/her valuable suggestions and constructive comments, which helped us in improving the quality of this manuscript. Please find attached a revised version showing changes that were made. Language and style have been improved. In addition, the manuscript has been proofread

Reviewer 3 Report
In general, the authors have addressed all of my suggestions and concerns in their manuscript revisions. I believe that the revised manuscript has improved significantly and is suitable for publication. However, I recommend further improvement of the English language in the new text that has been added in the revised version.
Author Response

(The authors gave the same response as above.)
